# Influence of Chitosan 0.2% in Various Final Cleaning Methods on the Bond Strength of Fiberglass Post to Intrarradicular Dentin

**DOI:** 10.3390/polym15224409

**Published:** 2023-11-15

**Authors:** Naira Geovana Camilo, Alex da Rocha Gonçalves, Larissa Pinzan Flauzino, Cristiane Martins Rodrigues Bernardes, Andreza Maria Fábio Aranha, Priscilla Cardoso Lazari-Carvalho, Marco Aurélio de Carvalho, Helder Fernandes de Oliveira

**Affiliations:** 1Department of Endodontics, School of Dentistry, Evangelical University of Goiás, Anápolis 75083-515, GO, Brazil; nairageovana15@gmail.com (N.G.C.); dralexdarocha@gmail.com (A.d.R.G.); cristiane.bernardes@unievangelica.edu.br (C.M.R.B.); 2Department of Oral Biology, School of Dentistry, University of Cuiabá, Cuiabá 78065-900, MT, Brazil; larissa_pinzan@hotmail.com (L.P.F.); andreza.maria@kroton.com.br (A.M.F.A.); 3Department of Restorative Sciences, School of Dentistry, Evangelical University of Goiás, Anápolis 75083-515, GO, Brazil; priscilla.lazari@docente.unievangelica.edu.br (P.C.L.-C.); marco.carvalho@docente.unievangelica.edu.br (M.A.d.C.)

**Keywords:** dentin-bonding agent, chelators, fiber post, ultrasonics

## Abstract

The purpose of this study was to analyze the influence of Chitosan 0.2% in various final cleaning methods on the bond strength of fiberglass post (FP) to intrarradicular dentin. Ninety bovine incisors were sectioned to obtain root remnants measuring 18 mm in length. The roots were divided: G1: EDTA 17%; G2: EDTA 17% + PUI; G3: EDTA 17% + EA; G4: EDTA 17% + XPF; G5: Chitosan 2%; G6: Chitosan 2% + PUI; G7: Chitosan 2% + EA; G8: Chitosan 2% +XPF. After carrying out the cleaning methods, the posts were installed, and the root was cleaved to generate two disks from each root third. Bond strength values (MPa) obtained from the micro push-out test data were assessed by using Kruskal–Wallis and Dwass–Steel–Critchlow–Fligner tests for multiple comparisons (α = 5%). Differences were observed in the cervical third between G1 and G8 (*p* = 0.038), G4 and G8 (*p* = 0.003), G6 and G8 (*p* = 0.049), and Control and G8 (*p* = 0.019). The final cleaning method influenced the adhesion strength of cemented FP to intrarradicular dentin. Chitosan 0.2% + XPF positively influenced adhesion strength, with the highest values in the cervical third.

## 1. Introduction

Endodontically managed teeth presenting a considerable coronal structure loss after endodontic treatment imposes the necessity of fiberglass posts (FP) associated with resin cements to restore the biomechanical form and function [1]. The recommendation of these materials is based on characteristics of durability, aesthetics, and operational cost, which make them a very favorable option for the restoration of the teeth in question [2]. However, it must be considered that during the preparation for the intraradical retainer, the formation smear residue resulting from the infected dentin debris, remains of gutta-percha and obturator cement can negatively influence the polymerization processes of resin cement and consequently the final cementation quality of the FP which is essential for successful adhesion [3].

The union of demineralized intrarradicular dentin with the cement and the FP is based on micromechanical retention and therefore, dentine cleaning of the walls of the intraradicular dentin is imperative for ideal retention of the post [4]. In this sense, the efficiency of endodontic irrigation in the sanitization and cleaning stages of the root canal becomes fundamental in reaching areas with difficult access that were untouched during the instrumentation stage, such as isthmuses and lateral canals [5]. Technological advances have allowed irrigating solutions to be agitated within the root canal for more effective smear layer clearance through mechanical, sonic, or ultrasonic agitation methods [6]. Among the solutions tested to remove this smear layer, EDTA and its combinations (mainly sodium hypochlorite) are the most used due to their chelating properties [7]. However, the prolonged use of EDTA can cause erosion in the dentin matrix, thus compromising the bond strength between the post and dentin [8,9]. Thus, there is a search for alternative solutions that are more biocompatible than EDTA in an effort to reduce these possible damages [7,10]. As an alternative, chitosan has been increasingly studied because it is a natural solution, is biocompatible with tissues, and has adequate properties of biodegradability, bioadhesion, and low cytotoxicity [11,12,13,14,15,16]. The chelating capacity of chitosan has attracted substantial interest due to its strong affinity for various metal ions in acidic pH environments [17,18]. This unique feature has expanded the potential of chitosan as a substitute for EDTA, which not only causes erosion but also poses environmental risks [19]. Previous studies have highlighted the favorable impact of chitosan in minimal concentrations and short-term applications on the demineralization of intraradicular dentin. Furthermore, it presents a cleaning capacity similar to other chelating agents used in clinical practice, such as citric acid and EDTA [7,10,11,17,19].

There is still a notorious absence of consensus about the real influence of post preparation cleaning procedures on the adhesion of FP to radicular dentin [4,20]. So far, there has been a lack of studies evaluating the influence of chitosan in association with various agitation methods, on the adhesion process, and on the biomechanical bond strength of FP. Therefore, it seems opportune to investigate the influence of the final cleaning method on the adhesive force between the post and luting agent and intrarradicular dentin. The null hypotheses assessed were that there would be no difference in the level of bond strength depending on (i) the chelating solution, (ii) the chelator activation method, and (iii) the third of the radicular canal.

## 2. Materials and Methods

The research was approved by the Animal Ethics Boards of the Universidade Evangélica de Goiás, Brazil (#001/2021). Three hundred extracted bovine lower incisors with fully developed roots, anatomically analogous in size and shape [21,22,23] were obtained and stored in 0.2% thymol solution (Fitofarma, Goiânia, GO, Brazil). Periapical radiographs were obtained to verify the samples’ normality, and only teeth with a unique root canal without obliterations were included in the study. In total, 90 samples were utilized.

### 2.1. Endodontic Instrumentation and Obturation

The crowns of the teeth were sectioned using a dual-sided diamond disc (KG Sorensen, Sao Paulo, SP, Brazil) perpendicular to its long axis obtaining standardized roots 18 mm in length from the apical end. A #15 K-file (Dentsply Maillefer, Ballaigues, Switzerland) was used to verify the patency of all root canals. 

The working length (WL) was established using a #15 K file (Dentsply Maillefer), which was introduced into the root canal until it was visible in the apical foramen. The WL was set 1 mm short of this measurement. To simulate clinical conditions, the root apexes were sealed with flow composite (Top Dam, Dental Products, São Paulo, SP, Brazil). ProTaper^®^ Gold instruments (Dentsply) were utilized for endodontic instrumentation. The channels were instrumented until reaching the instrument F5 (50/0.05). Each instrument was used in the instrumentation of just five root canals through X-Smart Plus endodontic motor (Dentsply), with speed and torque standards established by the manufacturer. During instrumentation, the canals were irrigated with 4 mL of 2.5% sodium hypochlorite (Fitofarma, Goiânia, GO, Brazil). The root canals were irrigated with 17% EDTA (Biodinamica, Ibiporã, PR, Brazil) for 3 min to remove the smear layer. 

The roots were subsequently dried using absorbent paper points (Dentsply, Charlotte, NC, USA) and then filled with gutta-percha cones and epoxy resin-based cement (AH Plus; Dentsply), mixed according to the instructions of the manufacturer using Tagger’s hybrid technique. The canal access was sealed with micro-hybrid composite resin (TPH Spectrum, Dentsply Brazil, São Paulo, SP, Brazil). All roots were stored at 37 °C and 100% humidity for 7 days to allow the cement to light cure.

### 2.2. Post-Space Preparation 

After the obturation, heated condensers (Paiva; SS White, Piscataway, NJ, USA) were used to remove the initial portion of the root canal filling mass. The conduits were prepared to a depth of 14 mm using Largo drills #3–5 (Dentsply Maillefer), corresponding to fiber posts of 1.5 mm in diameter (Reforpost #3; Angelus, Londrina, Brazil) [1,21]. Root canals were irrigated with 4 mL of 2.5% NaOCl after each drill change and dried with absorbent paper cones.

### 2.3. Experimental Groups

The samples were randomly distributed into eight experimental groups and a control group, in accordance with the chelating agent tested and the activation method (Figure 1).

### 2.4. Formulation of Chelating Solutions 

The solutions were formulated in a compounding pharmacy (Fitofarma, Troyan, Bulgaria) and were prepared with analytical grade reagents and water purified by a Reverse Osmosis system with Ultraviolet Light (Quimis, Diadema, SP, Brazil) with electrical conductivity lower than 1 μS mm -two. The pH of the solutions was determined with a digital pH meter (Analion, Ribeirão Preto, SP, Brazil). The 0.2% chitosan solution was prepared with 0.2 g of chitosan (ACROS Organics Gell, Belgium; degree of deacetylation >90%) in 100 mL of 1% acetic acid. The mixture was stirred using a magnetic stirrer at 100 °C in 200 rpm for 2 h [7,11].


**CNI**


A total volume of 4 mL of 2.5% NaOCl, 4 mL of each chelating agent, and another 4 mL of NaOCl was introduced into the root canals using a 5 mL disposable syringe (Ultradent, Tokyo, Japan) and a 29-gauge needle (NaviTip; Ultradent, Tokyo, Japan). The needle was inserted 1 mm short of the cementoenamel junction (CT) without coming into contact with the canal walls. Each chelating agent was allowed to remain in the canal for a duration of 3 min without undergoing any activation process.


**PUI**


PUI was conducted in 3 cycles of 20 s each with 2 mL of the solution per cycle. The solutions were passively activated using EMS PM 200 ultrasound (EMS, Nyon, Switzerland) and a E1-Irrisonic tip (Helse, Sao Paulo, Brazil) positioned 1 mm short of the WL, without touching the walls of the root canals, so that it vibrated freely. The ultrasonic unit was adjusted to 10% power following the manufacturer’s specifications for the use of the insert [9,24].


**EA**


Three activation cycles were performed as previously described. The solutions were activated with the EndoActivator system (Dentsply Maillefer) and a medium activator tip (25/0.04), which was inserted 1 mm from the WL for 20 s (each cycle with 2 mL of the solution) at 10,000 cycles per minute.


**XPF**


Three activation cycles were performed as previously described. The solutions were activated with the XP-Endo Finisher (25/0.00) instrument (FKG Dentaire, La Chaux-de-Fonds, Switzerland), which was inserted 1 mm short of the WL. The instrument operated at a speed of 800 rpm and torque of 1 Ncm. Slow and smooth movements of penetration and withdrawal were performed for 20 seconds (each cycle with 2 mL of the solution). The cleaning methods were completed, and the canals were washed with 4 mL of saline solution and dried with absorbent paper tips.

### 2.5. Fiber Post Cementation

After applying a thin layer of utilitarian wax on the external surfaces of the roots to prevent lateral polymerization resulting from the photoactivation of the cement, the post underwent a 15-second cleaning with 70% alcohol. Subsequently, the silane (Silane, Angelus) was applied for 1 minute using a micro brush (KG Sorensen, Sao Paulo, SP, Brazil). The self-adhesive resin cement (RelyX U200; 3M-ESPE, St Paul, MN, USA) was manipulated according to the manufacturer’s instructions and inserted into each root canal with the assistance of a lentulo spiral instrument (Dentsply Maillefer) and applied to the surface of the fiberglass post. The post was inserted into the canal with appropriate digital pressure, removing excess cement with a clean micro brush (KG Sorensen) after one minute.

Three minutes later, the cement was light-cured using a 1200 mW/cm^2^ intensity source (Radii-Cal; SDI, Bayswater, Australia) for 40 seconds on the cervical region, along the long axis of the root, and obliquely on the buccal and lingual surfaces, totaling 120 seconds per root. The dentin-cement-post interface was sealed with composite resin to ensure a hermetic seal of the root canal, ensuring the integrity and stability of the procedure.

### 2.6. Root Sectioning Procedure

In the meticulous process of root sectioning, each root underwent careful transverse cutting using a double-sided diamond disc (4” diameter × 0.012” thickness × 1/2”; Arbor, Extec, Enfield, CT, USA) mounted on a specialized hard tissue microtome (Isomet 1000, Buehler, Lake Bluff, IL, USA) set at a low speed (400 rpm), ensuring precision and accuracy. Throughout the procedure, a continuous flow of water provided effective cooling. From different segments of the root—cervical, middle, and apical—two 1 mm thick discs were carefully obtained, yielding a total of 6 discs per root. These dentin discs were precisely cut at distinct measurements: 11 and 12 mm from the root apex for the cervical region, 8 and 9 mm for the middle region, and 5 and 6 mm for the apical region, ensuring comprehensive representation of each third of the root. This standardized method of sectioning, utilizing advanced equipment and precise measurements, guaranteed the consistency and reliability of the obtained samples, forming the foundation for subsequent analyses and evaluation.

### 2.7. Micro Push-Out Mechanical Test (MPMT)

For conducting the micro push-out test, a specially designed apparatus was employed, crafted with stainless-steel metal bases measuring 3 cm in diameter [25,26,27]. These bases featured central holes of 2, 3.5, and 4.5 mm (as shown in Figure 2A), accompanied by load applicator tips measuring 1, 1.75, and 2 mm in diameter (as illustrated in Figure 2B). After positioning the set on the base of the mechanical testing machine (Microtensile OM150, Odeme Dental Research, Luzerna, Brazil) (Figure 2C) containing a 10 Kgf load cell, the discs were positioned in the hole of the metal base and the set was aligned to the tip load applicator (Figure 2D). They were then subjected to compression loading in the apex/crown direction at a speed of 0.5 mm/minute until failure occurred. The displacement force values were obtained in KgF which was transformed into Newton. The bond strength, in MPa, was calculated by dividing the force (N) by the area of the adhesive interface. The area of the adhesive interface was calculated by multiplying the height of the disc by the perimeter of the channel lumen, which was considered an ellipse: $A=h\times (\pi \left(3\left(a+b\right)-\sqrt{\left(3a+b\right)\left(a+3b\right)}\right)$ where *h* is the height of the disk, *a* is the largest radius, and *b* is the smallest radius.

### 2.8. Analysis of the Failure Mode by Optical Microscopy

After the mechanical test, each specimen was stored individually in Eppendorf-type microtubes with distilled water for later analysis of the fracture pattern by using 40× optical microscopy without any type of treatment or previous preparation. All samples were analyzed with the aid of an optical microscope (Carl Zeiss, META, Berlin, Germany). The images were processed with the help of the Zeiss LSM Image Browser software version 4.2. Photomicrographs were always obtained with the same increase for all specimens. Failure modes were classified into six categories: Mode 1—adhesive between the post and resin cement, Mode 2—adhesive between resin cement and intrarradicular dentine, Mode 3—mixed, between post, resin cement, and intrarradicular dentine, Mode 4—cohesive in cement, Mode 5—cohesive in the post, and Mode 6—cohesive in dentine (Figure 3).

### 2.9. Data Analysis

The data analysis was conducted using Jamovi 1.1.9 software (The Jamovi Project, 2019). Various cleaning methods were compared by examining the frequency of failure modes expressed as a percentage within each tested group. The bond strength values (in MPa) obtained from the micro push-out mechanical test underwent rigorous assessment through the Kruskal–Wallis test, followed by the Dwass–Steel–Critchlow–Fligner test for detailed multiple comparisons (α = 5%). Intra-examiner agreement was meticulously assessed using the kappa coefficient, applied to 10% of the sample, ensuring the reliability of the results.

## 3. Results

The kappa coefficient, standing at 0.86, demonstrated a high level of intra-examiner agreement. Table 1 shows the medians and interquartile ranges for the diverse groups examined.

During the group analysis, no significant differences were detected among the root thirds (*p* > 0.05). When comparing cleaning methods, variations surfaced in the cervical third between groups G1 and G8 (*p* = 0.038), G4 and G8 (*p* = 0.003), and G6 and G8 (*p* = 0.049), as well as Control and G8 (*p* = 0.019). In the middle third, although the group comparison initially indicated distinctions (*p* = 0.017), these distinctions vanished upon adjustment for multiple comparisons. The comparison between groups G1 and Control (*p* = 0.051) and G8 and Control (*p* = 0.053) was the closest to statistical significance. In the apical third, no notable differences emerged between groups (*p* > 0.05). The failure modes are detailed in Table 2. The predominant failure was Mode 2, constituting 68.3% of the cases, while Mode 6 accounted for 28.7%.

## 4. Discussion

In the current investigation, exploring the impact of 0.2% chitosan in various final cleaning methods on the adhesion FP to intrarradicular dentin produced convincing results. The study revealed significant disparities in bond strength between groups G8 and G1, as well as between groups G4 and G6, highlighting the substantial influence of the cleaning protocol on adhesion. These findings led to the partial rejection of previously raised null hypotheses, shedding new light on the intricate dynamics of the FP dentin-root bond.

The micro push-out mechanical test, a well-established and widely recognized method for assessing the bond strength between dentin and FP, has been extensively highlighted in prior research [8,20,25]. Its widespread use is attributed to its ability to ensure a more uniform distribution of stresses, minimizing data distortion and reducing the likelihood of premature failures [27,28]. One of its significant advantages lies in enabling the assessment of multiple specimens from the same root, a factor crucial for robust comparative analyses. Moreover, this technique facilitates the exploration of regional variations within the root thirds, providing results that closely mirror real-world clinical scenarios [1,24,29].

The decision to use bovine teeth was based on their easier availability in comparison with human teeth. Additionally, they offer improved standardization of both the teeth’s age and the root canal space [21,22,26]. The very similar characteristics between bovine and human teeth, especially in mechanical tests that evaluate the bond strength to dentin and enamel, provide a solid scientific basis, affirming the relevance and suitability of the chosen experimental model [26,30,31]. 

The choice of the 2.5% sodium hypochlorite solution before using the chelator was based on its suitability, having been previously tested in other studies [24,25,32,33] and proven to be less likely to affect the micromechanical properties of dentin. Higher concentrations were avoided, as these may interfere with experimental results. Previous studies have already demonstrated that increases in NaOCl concentration can cause significant changes in organic and inorganic components, resulting in lower micromechanical resistance due to degradation of the collagen matrix [33,34].

A wide range of materials has been made available on the market for cementing FP [8]. The resin cement used in the study has the capacity to adhere to the tooth structure by two mechanisms: the acidic monomers hybridize the dentin, and the resin chemically interacts with the hydroxyapatite [3,24,25,26]. Prior research has demonstrated that the presence of the smear layer, which forms during the intrarradicular preparation, can prevent the demineralization process promoted by the cement. This interference adversely affects the adhesive capacity of the cement, compromising its ability to form a strong and enduring bond [35]. 

EDTA 17% is an important chelator in removing the smear layer [7,10]. However, prolonged use can result in erosive effects on the dentin, leading to a reduction in its microhardness. This can potentially harm the periapical tissues surrounding the tooth [36]. Studies have focused on more biocompatible alternatives to minimize their interference with adhesive and restorative procedures [20,24,25]. 

Chitosan, derived from the deacetylation process of chitin found in crab and shrimp shells, is a chemical substance of significant interest in dental research. Its cost-effectiveness, biocompatibility, and minimal cytotoxicity have made it a focal point in the exploration of natural polysaccharides for dental applications [7,12,13,14,15,16,37]. In comparisons between cleaning methods, differences in values were found only in the cervical third, as reported in previous investigations [1,21,22]. 

The findings revealed significant variations in bond strength. The use of 0.2% chitosan + XPF as a final cleaning protocol positively influenced the bond strength of FP to intrarradicular dentin, with the highest values in the cervical third but with no differences for the other thirds (Table 1). Notably, the combination of 0.2% chitosan and XPF positively affected the adhesive strength, especially in the cervical region. It is essential to highlight that the cement’s ability to adhere to intrarradicular dentin is influenced by the chosen cleaning technique, as highlighted by numerous studies [11,24,25,26]. However, it is essential to take into account the fact that a higher concentration of the chelating agent can lead to increased demineralization of the dentin matrix, exposing the collagen and causing a series of inconveniences, including the reduction of microhardness and notably higher incidences of adhesion failures.

These findings align with a previous study [25] that compared the impact of 0.2% chitosan and 17% EDTA utilizing various techniques, including conventional irrigation and PUI. The study revealed higher values for group 6 in the cervical third of the root canal.

Failure modes were analyzed using the serial root sectioning method, from which two slices were obtained per root third, allowing direct inspection of the cementoenamel junction. As for the mode of adhesive failure, it was found that this most frequently occurred, followed by cohesive failure in the dentin (Table 2). This corroborates previous studies that demonstrated greater fragility at the cement-dentin interface [8,21,22,24,25], which can be justified by the presence of remaining obturators that adhered to the intraradicular dentin walls and within the dentinal tubules [4].

Certain limitations of the present study deserve attention in future research efforts. Notably, the samples were not exposed to mechanical and thermal conditions, which could better reproduce oral conditions, offering more authentic results [38]. Clinical trials are essential to confirm the results of this study and evaluate the effectiveness of new materials and methods for cleaning intrarradicular dentin post-retainer preparation, particularly in the context of FP-related restorations.

## 5. Conclusions

According to the methods used, it can be concluded that:1.the use of 0.2% chitosan + XPF as a final cleaning method positively influenced the bond strength between FP and intrarradicular dentin, with higher bond strength values in the cervical third.

## Figures and Tables

**Figure 1 polymers-15-04409-f001:**
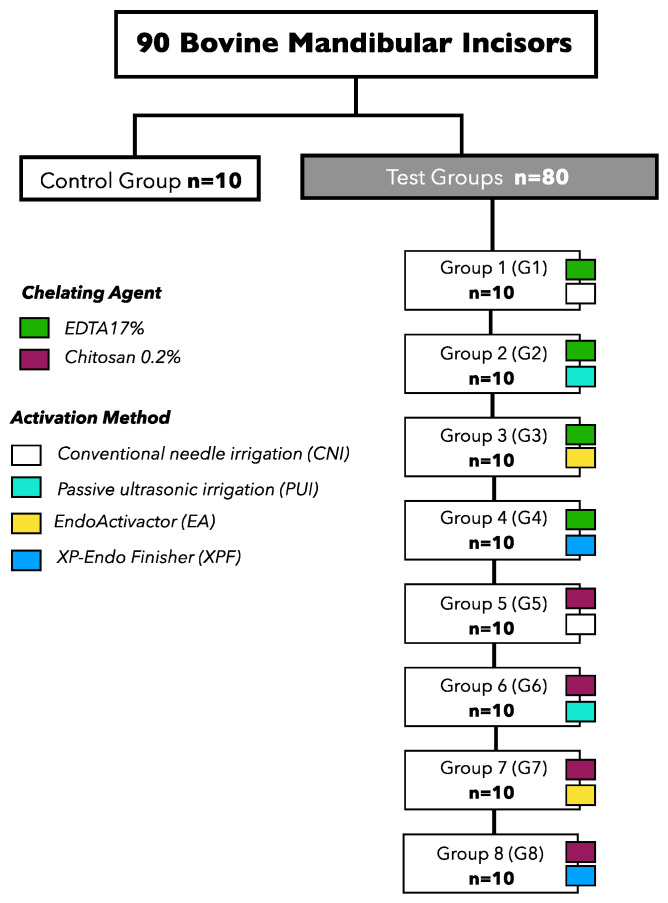
The allocation of experimental groups based on the chelating solution and activation method examined.

**Figure 2 polymers-15-04409-f002:**
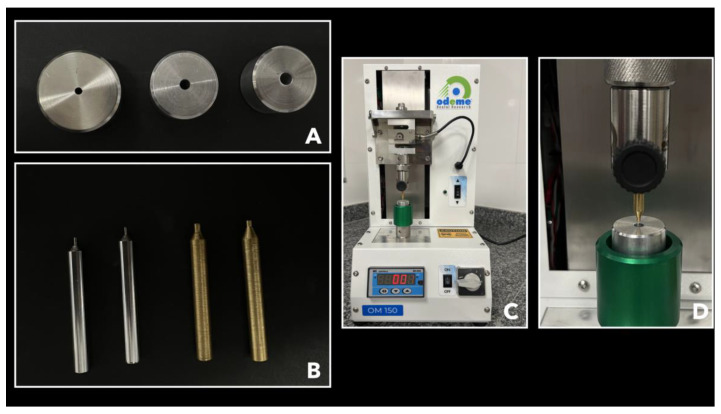
(**A**) Metal bases with 2, 3.5, and 4.5 mm diameter holes in the central region, (**B**) Applicator tips with 1, 1.75, and 2 mm diameter, (**C**) Positioning of the set on the base of the mechanical testing machine (Microtensile OM150, Odeme Dental Research, Brazil), (**D**) Discs aligned in a manner where the load applicator tip precisely matched the orifice in the metallic base.

**Figure 3 polymers-15-04409-f003:**
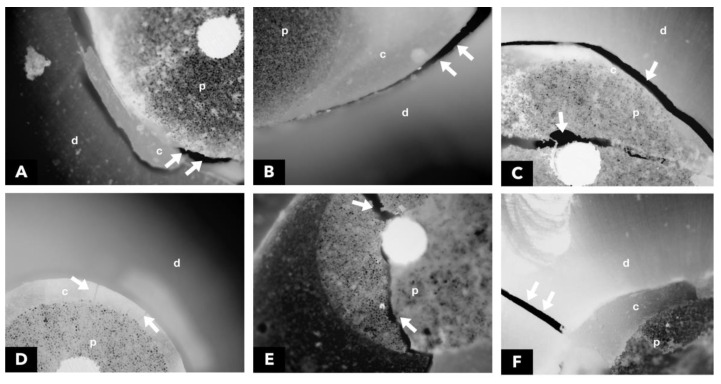
Failure Mode Types: (**A**) Mode 1, (**B**) Mode 2, (**C**) Mode 3, (**D**) Mode 4, (**E**) Mode 5 (**F**) Mode 6. Arrows point to the failure region. d-dentine, c-cement, and p-post.

**Table 1 polymers-15-04409-t001:** The median and interquartile range (IQR 25–75%), representing the bond strength values in the tested groups categorized by different root thirds.

Experimental Groups	Cervical Third		Middle Third		Apical Third		
	Median	IQR (25–75%)	Median	IQR (25–75%)	Median	IQR (25–75%)	*p*-Value
Control Group	2.48 ^A,a^	1.87–3.81	2.00 ^A^	1.58–3.05	2.72 ^A^	1.39–3.79	0.387
G1. EDTA 17%	2.54 ^A,a^	1.79–3.74	3.42 ^A,a^	2.89–4.06	3.88 ^A,a^	1.98–5.00	0.116
G2. EDTA 17% + PUI	3.79 ^A,a,b^	2.93–5.13	3.76 ^A,a^	1.98–5.39	3.43 ^A,a^	1.21–4.78	0.387
G3. EDTA 17% + EA	2.71 ^A,a,b^	1.80–3.50	2.74 ^A,a^	1.63–4.61	2.27 ^A,a^	0.94–4.76	1.000
G4. EDTA 17% + XPF	3.16 ^A,a^	2.65–3.64	2.46 ^A,a^	1.72–3.76	3.09 ^A,a^	1.88–3.62	0.796
G5. Chitosan 0.2%	2.90 ^A,a,b^	2.33–4.33	2.83 ^A,a^	1.91–4.13	2.81 ^A,a^	1.54–3.70	0.705
G6. Chitosan 0.2% + PUI	2.45 ^A,a^	1.19–3.49	2.79 ^A,a^	1.89–3.29	2.52 ^A,a^	1.66–3.53	0.212
G7. Chitosan 0.2%+ EA	3.91 ^A,a,b^	2.34.5.46	2.59 ^A,a^	1.22–3.45	3.01 ^A,a^	1.26–3.91	0.200
G8. Chitosan 0.2% + XPF	5.35 ^A,b^	3.13–6.14	4.01 ^A,a^	2.64–5.26	4.11 ^A,a^	2.79–4.90	0.350
*p*-Value	<0.01		0.017		0.159		

Different lowercase letters in the column indicate significant differences (*p* < 0.05). Different uppercase letters in lines indicate significant differences (*p* < 0.05).

**Table 2 polymers-15-04409-t002:** Percentage (%) of failure modes in the six different categories.

Experimental Groups	Failures Modes N (%)						
	1	2	3	4	5	6	TOTAL
Control Group	0 (0%)	41 (68.3%)	0 (0%)	0 (0%)	0 (0%)	19 (31.7%)	60 (100%)
G1. EDTA 17%	1 (1.7%)	31 (51.71%)	4 (6.71%)	1 (1.7%)	0 (0%)	23 (38.3%)	60 (100%)
G2. EDTA 17% + PUI	1 (1.7%)	41 (68.3%)	3 (5%)	0 (0%)	0 (0%)	15 (25%)	60 (100%)
G3. EDTA 17% + EA	0 (0%)	40 (66.7%)	0 (0%)	0 (0%)	0 (0%)	20 (33.3%)	60 (100%)
G4. EDTA 17% + XPF	0 (0%)	41 (68.3%)	1 (1.7%)	0 (0%)	0 (0%)	18 (30%)	60 (100%)
G5. Chitosan 0.2%	0 (0%)	35 (58.3%	4 (6.7%	1 (1.7%)	0 (0%)	20 (33.3%)	60 (100%)
G6. Chitosan 0.2% + PUI	0 (0%)	50 (83.3%	0 (0%)	0 (0%)	0 (0%)	10 (16.7%)	60 (100%)
G7. Chitosan 0.2% + EA	0 (0%)	40 (66.7%)	0 (0%)	0 (0%)	0 (0%)	20 (33.3%)	60 (100%)
G8. Chitosan 0.2% + XPF	0 (0%)	50 (83.3%)	0 (0%)	0 (0%)	0 (0%)	10 (16.7%)	60 (100%)
TOTAL	2 (0.4%)	369 (68.3%)	12 (2.2%)	2 (0.4%)	0 (0%)	155 (28.7%)	540 (100%)

## Data Availability

The data presented in this study are available on request from the corresponding author.

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
