# Peer review of "Influence of Chitosan 0.2% in Various Final Cleaning Methods on the Bond Strength of Fiberglass Post to Intrarradicular Dentin"

_polymers, 2023, doi:10.3390/polym15224409_

Round 1
Reviewer 1 Report
Comments and Suggestions for Authors
1. When passive ultrasonic flushing, what is the power of EMS PM 200 ultrasonic passive activation solution, if other power is used for ultrasound, whether it will affect the experimental results, if not, please provide relevant literature.
2. In the root slicing process, please indicate the specific speed of using the low speed setting.
3. Whether chitosan is more economically valuable than EDTA and has a wider range of use.
4. If the root canal is flushed with other concentrations of NaOCl, whether it will affect the experimental results, if not, please provide relevant literature.
5. In the process of chelate preparation, when stirring with a magnetic stirrer for 2h, please indicate how many revolutions the speed of the stirrer is.
6. In the process of chelate preparation, what is the temperature when stirring, please indicate.
7. Please increase the number of references appropriately.
8. Articles related to this topic in this journal can be added appropriately.
English language required to improve
Author Response
Dear Reviewer 1,
We appreciate all your considerations.
All answers can be found in the attached file and also in the revised manuscript with the change highlighted in red in the text.
Please check the attachments.
For further clarification, we are at your disposal
Kind regards
Prof. Helder Fernandes

Reviewer 2 Report
Comments and Suggestions for Authors
The following are areas that need correction.
In Figure 1, the name of the chelating agent in the legend should be consistent with that in the article. The word "quitosana" should be modified to "chitosan". '0,2%' should be modified to “0.2%”. Tables 1 and Table 2 have the same situation.
On page 4, line148. The unit of intensity is mW/cm2. Number 2 should be superscripted.
In Table 2, the number of failure modes #1 in the G2 group (EDTA 17% + PUI) group should be 1, not 0.
Reviewer comments
In the Introduction, the authors do not explain the reason for using chitosan as a chelating solution. Although the reference [11] has an explanation, it is recommended to add additional explanations.
The use of EDTA or 0.2% chitosan combined with different cleaning methods does have an impact on the structure of the smear layer. However, throughout the article, there is no surface morphology analysis before and after sample treatment, and the failure mode cannot be further discussed.
The results show that the G8 group (Quitosana 0.2% + XPF) group has a better bonding strength. Can you provide the reason?
Author Response
Dear Reviewer 2,
We appreciate all your considerations.
All answers can be found in the attached file and also in the revised manuscript with the change highlighted in red in the text.
Please check the attachments.
For further clarification, we are at your disposal
Kind regards

Reviewer 3 Report
Comments and Suggestions for Authors
The authors of this paper have evaluated how various final cleaning methods, particularly the inclusion of Chitosan 0.2%, impact the bond strength of fiberglass posts to intraradicular dentin.
The methodology is described in details and clearly indicates the root division into different groups and the specific cleaning protocols used. Why metal moulds were used for push out test, are there any other options to test the bond strength. Please provide refereence to your protocol.
The results highlights key differences in bond strength, particularly in the cervical third, between various groups. It underscores that the combination of Chitosan 0.2% with XPF (XPF) had a positive influence on adhesion strength, especially in the cervical third. Please discuss further Table 1 and 2 for any collration.
It might be beneficial to provide a bit more detail on the specific cleaning protocols used, as well as the role of Chitosan 0.2%, impact the bond strength of fiberglass posts to intraradicular dentin.
The methodology is described in details and clearly indicates the root division into different groups and the specific cleaning protocols used. Why metal moulds were used for push out test, are there any other options to test the bond strength. Please provide refereence to your protocol.
The results highlights key differences in bond strength, particularly in the cervical third, between various groups. It underscores that the combination of Chitosan 0.2% with XPF (XPF) had a positive influence on adhesion strength, especially in the cervical third. Please discuss further Table 1 and 2 for any collration.
It might be beneficial to provide a bit more detail on the specific cleaning protocols used, as well as the role of Chitosan and impact on bond strenght as this is a crucial aspect of the study. This additional information can help readers understand the technical aspects of the research more comprehensive as this is a crucial aspect of the study.
Author Response
Dear Reviewer 3,
We appreciate all your considerations.
All answers can be found in the attached file and also in the revised manuscript with the change highlighted in red in the text.
Please check the attachments.
For further clarification, we are at your disposal
Kind regards

Round 2
Reviewer 2 Report
Comments and Suggestions for Authors
1.In Figure 1, the name of the chelating agent in the legend should be consistent with that in the article. The word "quitosana" should be modified to "chitosan". '0,2%' should be modified to “0.2%”. Table 2 (G5 and G7 group)have the same situation.
Author Response
Dear reviewer, the text has been modified according to the scores.
See attached.
Main manuscript was also edited (attachment)
